# Feature of Nonlinear Electromagnetic Properties and Local Atomic Structure of Metals in Two Systems of Nanocomposites Co_x_(MgF_2_)_100−x_ and (CoFeZr)_x_(MgF_2_)_100−x_

**DOI:** 10.3390/nano15060463

**Published:** 2025-03-19

**Authors:** Evelina Pavlovna Domashevskaya, Sergey Alexandrovich Ivkov, Elena Alexandrovna Ganshina, Lyubov Vladimirovna Guda, Valeriy Grigoryevich Vlasenko, Alexander Victorovich Sitnikov

**Affiliations:** 1Department of Solid-State Physics and Nanostructures, Voronezh State University, Universitetskaya pl., 1, 394018 Voronezh, Russia; ivkov@phys.vsu.ru; 2Department of Magnetism, Faculty of Physics, Lomonosov Moscow State University, Leninskiye Gory B 1-2, GSP-1, 119991 Moscow, Russia; eagan@mail.ru; 3The Smart Materials Research Institute, South Federal University, St. Sorge, 5, 344090 Rostov-on-Don, Russia; astrosfedu@mail.ru (L.V.G.); vlasenko@sfedu.ru (V.G.V.); 4Department of Solid State Physics, Faculty of Radio Engineering and Electronics, Voronezh State Technical University, Moskovsky pr., 14, 394026 Voronezh, Russia; myftt@yandex.ru

**Keywords:** metal–dielectric and alloy–dielectric nanocomposites, nanocrystals, phase transformations, percolation threshold, nonlinear properties, soft and hard magnetic materials, K-edges XANES spectra of Co and Fe atoms

## Abstract

Based on modern concepts of the nonlinear percolation mechanisms of electrical and magnetic properties in granular metal–dielectric nanocomposites, the authors present for the first time a comparative analysis of their own results of a comprehensive study of nonlinear electromagnetic properties in two nanocomposite systems: metal–dielectric Co_x_(MgF_2_)_100−x_ and alloy–dielectric (CoFeZr)_x_(MgF_2_)_100−x_, obtained by ion-beam sputtering of composite targets in a wide range of different compositions. For the first time, the features of the influence of atomic composition and structural-phase transitions on nonlinear magnetoresistive, magnetic, and magneto-optical properties in two systems are presented in comparison, one of which, Co_x_(MgF_2_)_100−x_, showed soft magnetic properties, and the second, (CoFeZr)_x_(MgF_2_)_100−x_, hard magnetic properties, during the transition from the superparamagnetic to the ferromagnetic state. Moreover, for the first time, the concentration dependences of the oscillating fine structure of XANES K-absorption edges of Co atoms in the first system and Co and Fe atoms in the second system are presented, which undergo changes at the percolation thresholds in each of the two systems and thus confirm the nonlinear nature of the electromagnetic properties changes in each of the two systems at the atomic level.

## 1. Introduction

Metal–dielectric nanocomposites are granular materials with a complex nanometer-scale structure and magnetic metal granule sizes from units to tens of nanometers in a dielectric matrix. Using the compositional and dimensional features of nanocomposites, it is possible to regulate their various physical properties and use these materials in modern electronic devices [1,2,3,4,5,6,7]. An important parameter affecting the multifunctional properties of nanocomposites is the size of the metal granules, controlled by the conditions of production and the volume fraction of metal (x), which can vary from 0 to 1.0. In nanocomposites with a small x value, the metal granules are isolated from each other in the matrix volume, so such media are close in their properties to dielectrics. Conductivity in such composites is carried out mainly due to electron tunneling between metal granules or due to hopping conductivity along localized states in the dielectric matrix [8,9,10,11]. On the other hand, for compositions with a large x value, the metallic conductivity mode is realized. In this case, the size and number of granules per unit volume increase so much that conductive clusters and continuous metallic channels (chains or “nets” of granules in mutual contact with each other) are formed, penetrating the entire material and providing a predominantly metallic type of conductivity. Between the channels, there are dielectric regions that increase the overall level of electrical resistance of the material but do not affect the conductivity mechanism as a whole. In this mode, the material behaves as a metallic conductor, although those of its properties that depend on the mean free path of an electron are significantly changed due to strong scattering at the boundaries of the granules [1,2]. The electrical conductivity of such composites is several times lower than the values characteristic of pure metals or metal alloys. An extended conductive “net” and a magnetic closed structure arise in composites upon reaching the so-called percolation threshold. 

Thus, the percolation threshold x_per_ is the composition of the nanocomposite, at which the conductivity mechanism changes from tunnel or hopping to metallic due to the formation of conductive clusters and continuous conductive channels from metal granules. It has been experimentally established that for a large number of granulated composites, the percolation threshold is in the range of x_per_ ≈ 0.5–0.6 [11].

Due to the unique nanostructure, granular composites exhibit a wide variety of conductive, superconductive, optical, magnetic, mechanical, and other physical properties. Among them are giant magnetoresistance (GMR) [3,4,5,6,7,8], anomalous Hall effect [12,13], and nonlinear optical properties [14,15].

The magnetoresistive effect, or magnetoresistance (MR), consists of changing the specific electrical resistance of a material when it is placed in an external magnetic field. The absolute value of MR is determined according to the expression:(1)$$\Delta R/R_{0}=\left[(R_{H}-R_{0})/R_{0}\right]\cdot 100\%$$
where R_H_ is the electrical resistance of the material in the presence of an external magnetic field of strength H; R_0_ is the electrical resistance in a zero magnetic field. This phenomenon is observed in many conductive homogeneous media, metals, alloys, and semiconductors, but the absolute value of magnetoresistance does not exceed fractions of a percent. The nature of MR in homogeneous materials is ballistic and consists of the trajectory curvature of charge carriers under the influence of a magnetic field. As a result, the magnetoresistive effect is positive; that is, with an increase in the magnetic field strength, the electrical resistance of materials increases. Interest in the phenomenon of magnetoresistance and the magnitude of this effect itself changed radically when the level of technology development made it possible to obtain heterogeneous multiphase materials with a scale of inhomogeneities from units to tens of nanometers. In such materials, under certain conditions, negative MR is observed, sometimes reaching even several tens of percent. For such values of magnetoresistance, a special term, “giant magnetoresistance” (GMR), was introduced. Intensive studies of various materials containing 3d metals and possessing giant magnetoresistance began in the last century [1,2,3,4,5,6,7,8] but still arouse unabated interest in the scientific community [9,10,11,12,13,14,15]. Among such materials, a special place is occupied by nanogranulated metal–dielectric films [7,8,11], in which the GMR is the result of spin-dependent electron tunneling, i.e., it represents tunnel magnetoresistance (TMR) [7,8,9,10,11].

It was found that the magnetoresistance in granular nanocomposites is negative, and the idea was also expressed that the observed phenomenon is associated with the main mechanism of electrical conductivity in composites—with spin-dependent tunneling of electrons between neighboring granules [15,16]. At the same time, there are a sufficient number of works in which the TMS of various nanocomposites was discovered and studied [11,12,13,14,15,16,17].

To build quantitative models of TMR of composites, models are created to describe the properties of a single tunnel junction. Indeed, a nanogranulated composite can be represented as a set of a large number of single junctions connected to each other randomly. Slonczewski proposed a basic model of tunneling between two ferromagnetic electrodes [18]. This model considers charge transfer and, accordingly, electric current through a rectangular barrier separating ferromagnetic metals with almost free electrons. It is obvious that the GMR/TMR non-monotonically depends on the ratio of the metallic and dielectric components, since up to the percolation threshold the composite structure is a set of isolated metallic granules in a dielectric medium, while after the percolation threshold the composite consists of continuous metallic regions separated by small dielectric layers. According to a number of works [9,10,11], the concentration position of the maximum magnetic resistance in composites is located near the percolation threshold. This is primarily a consequence of the morphological features of these materials at threshold concentrations preceding the change in the conductivity mechanism. To consider the electron transport properties of such heterophase alloys, the percolation theory is used. In composites with a small content of the metallic component, the conductive granules are surrounded by a dielectric matrix, and the process of electrical transport is carried out due to the tunneling of electrons from granule to granule or due to thermally activated jumps along localized states [19,20,21]. The dielectric mode is characterized by high values of specific electrical resistance, increasing by several orders of magnitude with a decrease in the proportion of metal in the composite from 50% to 0. When cooling, the electrical resistance of the composites increases exponentially, and in the temperature range of 4.2–300 K, the change in ρ reaches several orders of magnitude, which indicates a thermally activated mechanism of electrical conductivity [15].

A basic model explaining the mechanism of electrical transport in granular metal–insulator composites below the percolation threshold was proposed by Sheng and Abeles [1,2]. The activated tunneling model assumes that charge transfer occurs via electron tunneling directly from one granule to another through dielectric barriers. Tunneling conductivity exponentially depends on the parameters of the barrier separating the metal granules:(2)$$\sigma \propto \exp(-2(2\pi /\hbar ){\left(2m\varphi \right)}^{1/2}s)$$
where h is Planck’s constant; m is the effective mass of an electron; ϕ is the effective barrier height, and s is the barrier width equal to the shortest distance between the granule boundaries. The key point is that the Sheng–Abeles model allows one to quite adequately explain the phenomenon of giant magnetoresistance observed in granular composites [11]. 

However, when considering the processes of electrical transport in granular nanocomposites, one cannot ignore the role of the dielectric matrix as a medium that also contributes to electron transport. The amorphous nature of the dielectric structure determines the existence of a large number of localized states in its forbidden zone [19,20,21], through which the so-called hopping (or Mott) conductivity can be realized [19]. It should be emphasized that hopping conductivity in amorphous dielectrics is a kind of electron tunneling from an occupied localized state to an unoccupied state without activation in a delocalized zone. At the same time, in different granular systems, differing in the materials of both the dielectric and metallic components, a common pattern is observed. In composites with a metal component concentration x not exceeding 35–40 at.%, the experimental dependences correspond to the Sheng–Abeles model (lnρ~T^−1/2^) [1,2]. Near the percolation threshold, the situation changes, and the experimental data are linearized over the entire temperature range in coordinates corresponding to hopping conductivity.

(lnρ~T^−1/4^). Thus, the assumption about the presence of hopping conductivity, or, in other words, electrical transport through the dielectric matrix in granular composites, has experimental confirmation [11].

The nature of magnetic properties in NC is even more complex [22,23]. Nanogranulated composite materials containing a ferromagnetic metal or an alloy of several elements as a metallic component are characterized by a certain set of magnetic properties. These properties can be divided into microscopic, relating to each granule separately, and macroscopic, or corporate, relating to the entire composite as a whole. From the point of view of microscopic properties, a nanoparticle formed from a ferromagnetic material should be ferromagnetic below the Curie temperature. Indeed, it has been theoretically shown [22,23] that the only limitation for the occurrence of ferromagnetic ordering in a nanogranule is the size factor. A massive ferromagnetic material in the absence of external magnetic fields always spontaneously breaks up into domains, which is dictated by energy considerations and the desire of the system to exist with minimal energy. Unlike massive objects, ferromagnetic nanoparticles become single-domain even at zero external magnetic field [23,24]. From the point of view of the minimum free energy, the main reason for the formation of single-domain granules is the increase in the specific gravity of the surface energy of the boundary layers between domains with a decrease in the particle size. The energy of the domain wall becomes comparable (or greater) with the volume energy of the granule’s own magnetic field (i.e., the uncompensated external field of the magnetic particle), devoid of the structure of regions with a closed flow. Thus, as a result of a decrease in the size of the ferromagnetic granule, a moment comes when its entire volume is occupied by one domain, i.e., a single-domain state arises [23,24]. The exact values of the sizes at which the particles become single-domain depend on the shape (degree of non-sphericity), the properties of the elements that form the particles, and a number of other parameters. In general, the critical sizes for the transition to the single-domain state are within about 10–100 nm [23,24]. For most well-known metals, such as Fe, Co, Au, Cu, etc., granules can have a size from one to several tens of nanometers. This is the size range in which granules become single-domain, and a transition to the superparamagnetic state is possible. Despite the fact that granules in composites are ferromagnetic, macroscopically the composite is not ferromagnetic, since the magnetic moments of the granules are randomly oriented relative to each other in the absence of an external magnetic field. In this case, the macroscopic magnetization of the ensemble of nanoparticles will tend to zero. Since the behavior of such a material is formally similar to a paramagnet, in which the magnetic moment of a granule is considered instead of the magnetic moment of an atom, a special term was introduced to characterize such a material: superparamagnet [11].

The superparamagnetic state is typical for systems consisting of nanosized magnetic particles placed in a non-magnetic medium called a matrix. Superparamagnetic properties were also observed in ferromagnetic nanogranules distributed in a polymer matrix [25] or simply in thin films deposited on a non-conducting substrate [26]. In all cases, the classical condition for the existence of superparamagnetism is the absence of interaction between ferromagnetic granules.

The modern point of view is that when considering the magnetic properties of composites containing ferromagnetic nanosized particles, the possibility of interaction between the granules should be taken into account. Firstly, the exchange interaction arises between two ferromagnets separated by a non-magnetic dielectric barrier. A model of such interaction, caused by electrons tunneling from granule to granule, was proposed by Slonczewski [18].

Secondly, when considering a system of metal granules with magnetic moments and spatially separated by some distance relative to each other, one cannot help but take into account the dipole–dipole interaction [11], which is dominant in the case of nanocomposites, and with an increase in the proportion of the metal component, this interaction increases. The manifestation and influence of these interactions on the magnetic properties of composites also depend on the elemental composition of the metal and dielectric components.

However, regardless of the elemental composition of the dielectric or metal components, the magnetization curves of the composites exhibit general patterns: the magnetization curves of nanocomposites with a metal content up to the percolation threshold show no hysteresis, as well as no magnetic saturation in fields ~11 kOe. These facts indicate that at room temperature, pre-percolation metal–dielectric nanocomposite systems (x < x_per_) are superparamagnets SPM [27,28,29,30].

In the region of high concentrations of the metal component, beyond the percolation threshold x > x_per_, nanocomposites exhibit ferromagnetic (FM) properties. However, the magnetic percolation threshold x_fm_ may not coincide with the electrical transfer threshold x_per_ [16]; it is always slightly lower than x_fm_ ≤ x_per_, since ferromagnetic order can arise without physical contact between metal granules. Therefore, the magnetic percolation threshold x_fm_ can be defined as the concentration at which the magnetization and coercivity increase sharply.

In [31,32], it was shown that magneto-optical methods make it possible to detect the presence of both FM and SPM states in nanocomposites. The magneto-optical (MO) properties of the nanocomposites can be studied by the transversal Kerr effect (TKE) method, which consists of measuring the relative intensity of p-polarized light reflected by a sample upon magnetization in a magnetic field parallel to the sample surface and perpendicular to the plane of incidence of light. The TKE values are determined by the relation:δ = [I(H) − I(0)]/2I(0)(3)
where I(H) and I(0) are the intensities of reflected light in the presence and absence of a magnetic field, respectively.

Granular metal–dielectric nanocomposites can be obtained by various methods, but the most universal is ion sputtering [11]. The formation of a granular structure occurs on the surface of the substrate, where atoms or atomic complexes knocked out of the target are deposited. The separation of the condensing medium into two components—dielectric and metallic—is achieved as a result of the self-organization process, the driving force of which is the tendency to decrease entropy during the implementation of a non-stationary process, which is condensation from the gas phase [11].

The most commonly used matrices for granular composites are oxide dielectrics SiO_2_, Al_2_O_3_, MgO, HfO_2_, ZrO_2_, etc. Most metals used in these matrices, Fe, Co, Au, Cu, etc., form granules ranging in size from one to several tens of nanometers. In addition, technologies for producing composites with oxygen-free dielectrics have recently been implemented. For example, in [33], when studying the magnetotransport properties in (Fe_51_Co_49_)_32_(MgF_2_)_68_ films with Fe_51_Co_49_ alloy distributed in a MgF_2_ dielectric matrix, a giant magnetoresistance of 13.3% at 10 kOe at room temperature was discovered. In [34], the frequency dependence of the tunnel-type magnetodielectric effect in superparamagnetic nanostructures Co_x_(MgF_2_)_1−x_ was demonstrated with a precise change in x in the range from 0.06 to 0.2. In [35,36], the electrical and magnetoresistive properties of thin Co_x_(MgF_2_)_1−x_ films were investigated in a wide range of metal phase concentrations (14 ≤ x, at.% ≤ 62) in the initial state and after thermal annealing in vacuum. The percolation threshold for this system was set in the region of x = 30–36 at.% Co. The magnetoresistive effect of the studied samples reached 7% in a field of 10 kOe at a cobalt concentration of x = 25 at.%. When heating the samples to 250 °C, the magnetoresistance value increased in composites containing cobalt up to the percolation threshold and decreased in composites above the percolation threshold. It was found that the MgF_2_ matrix is resistant to thermal heating up to 250 °C, while increasing the temperature to 350 °C leads to the disappearance of the magnetoresistive effect.

Thus, by now a large number of original works and monographs have accumulated, which present various nonlinear effects in granular metal–dielectric composites.

*The aim of our work* is to show how the patterns of influence of composition and phase formation that we discovered manifest themselves not only in the nonlinear electromagnetic properties of film nanocomposites (NC) metal–dielectric of two poorly studied before us systems Co_x_(MgF_2_)_100−x_ and (CoFeZr)_x(_MgF_2_)_100−x_, but also in the fine structure of X-ray absorption near edge structure (XANES) spectra of cobalt and iron Co K-edge and Fe K-edge, i.e., at the atomic level.

## 2. Objects and Methods of Research

Composite films with different contents of metal Co or alloy CoFeZr in a dielectric matrix MgF_2_ were obtained using the original ion-beam deposition setup described in [11]. The composite target consisted of a massive plate of metal Co or metal alloy Co_45_Fe_45_Zr_10_ with uneven and asymmetric placement of dielectric inserts MgF_2_. As a result of using such a target in the sputtered material in one cycle in an argon atmosphere at an operating pressure of ~5 × 10^−4^ Torr on substrates, a given gradient of composite concentration is formed [11].

The layer thicknesses of film samples of granulated composites with different compositions smoothly changed from one edge of the substrate to the other in the range of 2–4 μm with an increase in the metal or alloy concentration x in the nanocomposite. The layer thicknesses were measured using a scanning electron microscope (SEM) of JEOL JSM-6380LV (Tokyo, Japan).

The concentrations of chemical elements included in the composites were measured by electron probe X-ray spectral microanalysis using an attachment to the SEM equipped with three crystal diffraction spectrometers and an energy dispersive analysis system, with an error of less than 1.0 at.% of the measured element content. The determined composition of the samples was the percentage of each chemical element in the (CoFeZr)_x_ (MgF_2_)_100−x_ composite layer, expressed in atomic percent. In this case, the variable value x of the metal component in the composite is the sum of the atomic concentrations of three alloy metals: Co, Fe, and Zr.

The phase composition of nanocomposites and the sizes of nanocrystals formed as a result of the self-organization of two components, metallic and dielectric in different ratios, were studied by X-ray diffraction (XRD) on a DRON-4 diffractometer (St. Petersburg, Russia) using Co Kα radiation. X-ray diffraction patterns were obtained in a step-by-step scanning model.

The tunneling magnetoresistance (TMR) in accordance with the Equation (1) was studied by directly measuring the electrical resistance of samples when changing an external constant magnetic field using the four-probe method on an ECOPIA HMS-3000 setup (Gyeonggi-do, South Korea).

The device kit included a certified magnetic attachment with a magnetic induction of B = 5.5 kG.

The magnetic properties of the samples were studied using a Lake Shore Vibrating Sample Magnetometer (VSM) 7407 (LakeShore Cryotronics, Inc., USA), in magnetic fields up to 16 kOe at T = 300 K.

The magneto-optical (MO) properties of the nanocomposites were studied in the transversal Kerr effect (TKE) geometry, which consists of a change in the intensity of p-polarized light reflected by a sample upon magnetization in a magnetic field parallel to the sample surface and perpendicular to the plane of incidence of light [37,38]. The TKE values are determined by Equation (3) and measured dynamically with magnetic field modulation. This method allows the use of a differential measurement scheme, ensuring an accuracy of measuring relative changes in the intensity of reflected light of ~10^−5^.

The TKE spectra were recorded in the energy range E = 0.5–4.0 eV with an applied magnetic field of up to 3.0 kOe and room temperature. The magnetic field dependences of TKE(H) were measured at three fixed energies to analyze the magnetic structure of the nanocomposites.

The X-ray absorption spectra of Co and Fe K-edge XANES were measured using a Rigaku R-XAS laboratory spectrometer (Tokyo Boeki, Japan). Figure 1 shows its main parts.

Monochromatic radiation from the anode of the X-ray tube operating at 18 kV and 60 mA was obtained after reflection from a Ge (311) single crystal bent according to Johansson (Johansson bent [10.1016/j.ccr.2020.213466]) [39]. The energy resolution is 1.3 eV in the X-ray energy range of 7000–8000 eV. An ionization chamber filled with argon (gas pressure 300 mbar) is used to determine the intensity of incoming X-rays. The signal from the film sample on the substrate was recorded in the fluorescence mode using an Amptek XR-100KR semiconductor silicon drift detector. For each studied sample with a certain x value, ten XANES spectra were measured and averaged. Background subtraction, normalization, energy equalization, and extraction of the oscillation function χ(k) were performed in the Athena program from the Demeter1-2 package [40].

## 3. Structural-Phase Transformations and Nonlinear Electromagnetic Properties of Variable-Composition Nanocomposites Co_x_(MgF_2_)_100−x_ and (CoFeZr)_x_(MgF_2_)_100−x_

The phase composition of the composites on the substrate is formed as a result of self-organization from two components, metallic and dielectric, in different ratios from one edge of the substrate to the other, depending on the location of these edges relative to the composite target with non-equidistantly located dielectric inserts [11]. After deposition of the composite, which smoothly changes the metal concentration x in the dielectric matrix and the thickness of the deposited layer over the substrate within a few microns, the substrate is cut into narrow strips about 3 mm wide and about 30 mm long to determine the atomic composition, structure, and further studies of the functional properties depending on the phase composition and atomic structure of the samples.

### 3.1. Metal–Dielectric Nanocomposites Co_x_(MgF_2_)_100−x_

First, we studied in detail the features of the atomic structure, phase formation, and substructure of thin-film composites Co_x_(MgF_2_)_100−x_ with metallic cobalt in the dielectric matrix MgF_2_ with a change of metal concentrations in the range of 14 ≤ x ≤ 63 at.% and showed [41] that at a cobalt content of x < 37 at.%, the metal Co is in an X-ray amorphous state in the form of clusters in the nanocrystalline matrix of MgF_2_ (Figure 2). At x = 37 at.%, a nonequilibrium state occurs, in which both components, the metal and the dielectric matrix of the film composite on the glass substrate, are in an X-ray amorphous state. But with an increase in the cobalt content to x = 42 at.% in the amorphous dielectric matrix MgF_2_, cobalt nanocrystals of hexagonal syngony with dimensions of about 10 nm are formed, predominantly oriented in the [001] direction of the α-Co hexagonal unit cell (Figure 2).

Then, we investigated the effect of the atomic composition and structural-phase state of Co_x_(MgF_2_)_100−x_ nanocomposites on their nonlinear transport, magnetic, and magneto-optical properties [42].

The concentration dependence of electrical resistance showed (Figure 3A) that at a low cobalt content in the range of x = 16–27 at.%, the resistance of nanocomposites reaches values of 10^7^–10^6^ Ohm, characteristic of traditional dielectrics, whereas in the range of x = 47–59 at.%, the resistance drops to values of the order of 1 Ohm or less, typical for metals. In the transition range x = 32–42 at.%, corresponding to the formation of cobalt nanocrystals and the appearance of contacts between them at the percolation threshold at x_per_ = 37 at.%, a sharp drop in resistance by 4–5 orders of magnitude occurs.

The concentration dependence of the tunnel magnetoresistance (TMR) of the samples of the Co_x_(MgF_2_)_100−x_ system in Figure 3B shows a particularly nonlinear character. To obtain this dependence, the value of TMR = ΔR/R(0) in accordance with Equation (1) was measured in a magnetic field with induction B = 5.5 kGs on 10 samples with different values of cobalt concentration, of which 5 samples up to the percolation threshold and with the maximum TMR contained amorphous cobalt in the nanocrystalline matrix MgF_2_, and 5 other samples beyond the percolation threshold x ≥ 37 contained hexagonal cobalt nanocrystals α-Co in the amorphous matrix MgF_2_. At a cobalt concentration of x = 27 at.%, up to the percolation threshold, there is a sharp increase in TMS to the maximum value TMR_max_ = ΔR/R(0) = 5%. It significantly exceeds the corresponding value of TMR_max_ in cobalt nanocomposites with other matrices: in the Co_x_(Al_2_O_3_)_100−x_ system TMR_max_ ≈ 1.4%, and in the Co_x_(SiO_2_)_100−x_ system TMR_max_ ≈ 0.5% in a magnetic field with an induction of 10 kGs [10].

The concentration dependence of the magneto-optical Kerr effect in equatorial geometry, the so-called transversal Kerr effect (TKE), which consists of changing the intensity of linearly polarized light reflected by a sample magnetized perpendicular to the plane of incidence of light, also turned out to be nonlinear. The ratio of the difference in the intensities of light reflected by a sample in magnetized I(H) and demagnetized I(0) states to the intensity of light I(0) determines the magnitude and sign of the TKE in accordance with the Equation (3).

It was found [42] that reaching of the percolation thresholds electric x_per_ and magnetic x_FM_ in Co_x_(MgF_2_)_100−x_ nanocomposites at x = 37 at.% is manifested in the form of a maximum of the absolute value on the concentration dependences of TKE at photon energy values of 1.97 eV and 3.28 eV (Figure 4A).

It is known that, according to the magnitude of the coercive force Hc, magnetic materials are divided into hard magnetic (Hc ≥ 50 Oe) and soft magnetic (Hc < 50 Oe). The concentration dependence of the coercive force Hc in the Co_x_(MgF_2_)_100−x_ nanocomposites in Figure 4B shows that at the percolation threshold at x_FM_ = 37 at.%, the system transitions from the superparamagnetic to the soft ferromagnetic state, which, with an increase in the cobalt content (x > 40 at.%), transforms into a hard magnetic state with a coercive force of Hc ≈ 95 Oe [43].

For comparison with our results, in the work [43], hexagonal Co nanocrystals, giving exactly the same three reflections on the diffraction pattern (100), (002), and (101), as in our Figure 2, were obtained by high-frequency magnetron sputtering on a glass substrate made of Co_85_Zr_15_ alloy in an Ar + O_2_ atmosphere. The atomic composition of the nanocomposite layer (about 600 nm thick) Co_66.5_Zr_6.8_O_25.7_, together with the XRD data, shows that as a result of self-organization under certain technological and thermodynamic conditions, a nanocomposite Co_66.5_(ZrO_2_ + xO_2_)_33.5_ is formed from Co nanocrystals in an amorphous dielectric matrix (ZrO_2_ + xO_2_), which exhibits soft magnetic properties, in contrast to our system of nanocomposites Co_x_(MgF_2_)_100−x_, which begins to exhibit hard magnetic properties at x > 40 at.%.

Having studied the regularities of nonlinear changes in phase-structural and electromagnetic properties depending on the ratio of metal and dielectric in the composition of the Co_x_(MgF_2_)_100−x_ system, we moved on to the study of nanocomposites of a more complex atomic composition (CoFeZr)_x_(MgF_2_)_100−x_ with a three-element magnetic alloy CoFeZr in the same oxygen-free dielectric matrix MgF_2_.Having studied the regularities of nonlinear changes in phase-structural and electromagnetic properties from the metal–dielectric ratio in the composition of nanocomposites of the Co_x_(MgF_2_)_100−x_ system, we moved on to the study of nanocomposites of a more complex atomic composition with a three-element magnetic alloy CoFeZr in the same oxygen-free dielectric matrix of MgF_2_.

### 3.2. Alloy–Dielectric Nanocomposites (CoFeZr)_x_(MgF_2_)_100−x_

It should be noted that in all the works conducted before us, the metallic component of the composites was obtained by sputtering targets made of polycrystalline metals Co, Fe, or CoFe alloy [3,4,5,6,7,8,9,10,33,34], while in our studies we use a target made of amorphous alloy Co_45_Fe_45_Zr_10_ [44] with inserts made of polycrystalline MgF_2_. It turned out that the use of an amorphous alloy instead of crystalline metals significantly affects the nature of interatomic interactions, phase transitions, and electromagnetic properties in such composites due to the significantly smaller sizes of metal clusters in the amorphous phase.

As described in the previous Section 3.1, using the example of a less complex metal–dielectric system Co_x_(MgF_2_)_100−x_, we showed [40] that at certain ratios of metal and dielectric concentrations in the nanocomposite, simultaneously with the direct transition of metallic Co from the amorphous to the nanocrystalline state, a mutually inverse (antibatic) phase transition of the dielectric phase MgF_2_ from the nanocrystalline to the amorphous state occurs. These two mutually inverse transitions are separated by a nonequilibrium state at x = 32–37%, in which the entire nanocomposite system is amorphous. But below we will show that the concentration dependences of phase transformations in the alloy–dielectric system (CoFeZr)_x_(MgF_2_)_100−x_ turn out to be even more complex and interesting.

Studies of variable composition nanocomposites (CoFeZr)_x_(MgF_2_)_100−x_ in the alloy concentration range x = 9–51 (at.%) using X-ray diffraction (XRD), X-ray photoelectron spectroscopy (XPS), and infrared spectroscopy (IR) have shown that the relative content of the metallic alloy CoFeZr in the oxygen-free nanocomposite has the most significant effect on its atomic structure, substructure [44], conductivity, and magneto-optical properties [45].

In Figure 5, we reproduce our original diffractograms of the nanocomposites with different x concentrations of CoFeZr alloy, which were first published in our work [44] with the samples on glass and sitall substrates and then reproduced in a subsequent work [45] with the samples only on glass substrates. As shown in Figure 5 from the work [45], at alloy concentrations x < 30 at.% it is in an X-ray amorphous state in the form of CoFeZr small clusters in the nanocrystalline matrix of MgF_2_. But already at x = 30 at.% the dielectric matrix of MgF_2_ undergoes an antibate transformation from the nanocrystalline to the amorphous state, and a nonequilibrium state occurs in which both components, the alloy and the matrix of the film composite on a glass substrate, are in an X-ray amorphous state. However, with an increase in the alloy content to x = 34 at.%, the first phase transition occurs when CoFeZr nanocrystals of hexagonal syngony with dimensions of about 10 nm are generated in the amorphous dielectric matrix of MgF_2_, predominantly oriented in the basal plane of the hexagonal unit cell (001) α-Co (Figure 5). The second phase transition from the hexagonal structure hcp α-Co to the body-centered cubic bcc α-Fe structure of CoFeZr nanocrystals occurs in the amorphous dielectric matrix of MgF_2_ with an increase in the relative content of the alloy to x = 43–47 at.% (Figure 5).

Apparently, phase transformations of this kind (hcp → bcc) in metallic nanocrystals, depending on the concentration x, are observed for the first time in the system (CoFeZr)_x_(MgF_2_)_100−x_ due to the presence of cobalt and iron in equal proportions in the initial amorphous target alloy Co_45_Fe_45_Zr_10_ with dielectric inserts of polycrystalline MgF_2._

As for the general situation regarding the studies of phase transitions in nanocomposites, they are very labor intensive and therefore few in number. However, we found a paper [46], in which, when studying Co-C nanocomposites using precision differential scanning calorimetry (DSC), the authors established polymorphic transformations in cobalt nanocrystals (α-Co hcp → β-Co fcc) from the low-temperature to the high-temperature phase in the temperature range from 380 to 480 °C, which is 100 °C lower than the corresponding range of the polymorphic transition α → β for metallic cobalt plates several mm thick [47]. However, the concentration phase transition of CoFeZr nanocrystals (α-Co hcp → α-Fe bcc) that we discovered at a constant temperature has a completely different nature compared to the temperature polymorphic transformations of metals and compounds of a certain composition and requires further research on other similar systems of nanocomposites of variable composition.

Our studies using the XPS method [44] with the use of ion etching of the surface layers of nanocomposites confirmed the presence of a predominant type of metallic bonds in the clusters and nanocrystals of CoFeZr, as well as ionic bonds in the dielectric matrix of MgF_2_, preserved as a result of self-organization under nonequilibrium conditions of ion-plasma sputtering. In this case, the main role of zirconium atoms of the sputtered three-element alloy Co_45_Fe_45_Zr_10_ is to bind impurity oxygen atoms with the formation of stable chemical bonds Zr-O with the formation of impurities ZrO_2_.

The studies of IR spectra in the same work [44] showed that changes in the sizes of nanocrystals and interatomic distances of the MgF_2_ matrix, occurring with an increase in the content of the metal alloy in (CoFeZr)_x_(MgF_2_)_100−x_ nanocomposites in the range of x = 7–25 at.% up to the percolation threshold, are accompanied by a narrowing of the width of the main band of the IR spectra and small changes in the position of the maximum towards higher frequencies (~500 cm^−1^) compared to polycrystalline MgF_2_ (~450 cm^−1^) [48].

At the same time, none of the methods we used to study interatomic interactions (XRD, XPS, IR) in [44] detected possible chemical bonds of the metal atoms with the matrix fluorine atoms at the metal–dielectric interphase boundaries.

As an example of such uncertainty, we present Figure 6 from our work [44], which shows the XPS spectra of fluorine F1s in the samples (CoFeZr)_x_(MgF_2_)_100−x_ with an alloy concentration of x = 9 (before the percolation threshold) and a concentration of x = 48 (after the percolation threshold), obtained at different ion etching times from a minimum of 30 s to a maximum of 20 min, after which the spectra practically do not change their shape. The results show that all XPS spectra of fluorine F1s are a single-component symmetric curve with a maximum binding energy of ~686.0 eV, corresponding to magnesium difluoride MgF_2_ [49] And only after deep etching for 10–20 min does a very weak component with a binding energy of about 685.0 eV appear in the spectra, which can relate to the boundary bonds of fluorine with any of the d-metal atoms Co, Fe, and Zr [49]. Therefore, in the next work [45], we attempted to detect the most probable interaction of iron and fluorine atoms using nuclear Mössbauer spectroscopy, which was crowned with success.

In Figure 7, we present Mössbauer spectra from the calibration sample of α-Fe (left) and from the nanocomposite sample (CoFeZr)_51_(MgF_2_)_49_, measured at room temperature (right) from our work [45].

The total amount of accumulated statistics in the Mössbauer spectrum from the (CoFeZr)_51_(MgF_2_)_49_ sample with the maximum magnetic alloy content x = 51 at.% was more than 500 h. In the spectrum of this sample, reproduced on the right side of Figure 6, an intense magnetic sextet (hyperfine magnetic splitting) stands out, which we attribute to iron in the composition of nanocrystals of the CoFeZr alloy with the bcc structure of α-Fe. However, along with the magnetic sextet, a paramagnetic quadrupole doublet stands out in the center of the spectrum of the nanocomposite, which belongs to iron fluoride FeF_2_. The ratio of the areas of these two spectra from two different phases in the nanocomposite sample magnetic alloy CoFeZr and paramagnetic phase FeF_2_ is 82:18 with a total area of 100 for the sample (CoFeZr)_51_(MgF_2_)_49_.

Thus, the appearance of an additional doublet from the paramagnetic phase FeF_2_ in the Mössbauer spectrum of the (CoFeZr)_51_(MgF_2_)_49_ nanocomposite indicated that chemical interaction of the boundary iron atoms with the fluorine atoms of the matrix occurs at the interfaces of the magnetic nanocrystals CoFeZr and the matrix MgF_2_, resulting in the formation of the paramagnetic phase of iron fluoride FeF_2_. This circumstance significantly affects the nonlinear electromagnetic properties of this system compared to the previous less complex system of nanocomposites Co_x_(MgF_2_)_100−x_ containing one metal Co in the same dielectric matrix, as we will show below.

Further, in Figure 8A we reproduce from our work [45e the concentration dependence of electrical resistance for samples of the (CoFeZr)_x_(MgF_2_)_100−x_ system with two extrapolation lines at different tilt angles. The intersection of these lines in the region x = 30–34 at.% corresponds to the electrical percolation threshold x_per_. Two straight lines of different slopes indicate different mechanisms of current flow before and after the percolation threshold along high-resistance and low-resistance paths.

Figure 8B reproduces the concentration dependence of the tunnel magnetoresistance TMR = ΔR/R(0) in accordance with Equation (1), which was measured in a magnetic field with induction B = 5.5 kGs on 11 samples with different concentrations of the CoFeZr alloy, of which 5 samples up to the percolation threshold and at the maximum TMR contained an amorphous alloy in a nanocrystalline MgF_2_ matrix, and 6 other samples beyond the percolation threshold x ≥ 34 at.% contained hexagonal nanocrystals of the alloy in an amorphous MgF_2_ matrix. At an alloy concentration of x = 25 at.%, up to the percolation threshold, a sharp increase in TMR occurs to the maximum value TMR_max_ = ΔR/R(0) = 2.4%. This value is less than the corresponding value of TMR_max_ in a composite of the same alloy in an oxide matrix (CoFeZr)_x_(Al_2_O_3_)_100−x_ with a value of TMC_max_ ≈ 3.5% in a magnetic field with an induction of 10 kGs [10]. Thus, the obtained concentration dependence of TMR is very non-monotonic: in the dielectric region of the composites, small TMR values are recorded, which sharply increase with the concentration of the metal component in the range of 15 < x < 27 to the maximum value of TMR_max_ = 2.4% at x = 25 at.%, and then at x > 27 at.% they also sharply decrease and remain insignificant in the low-resistance region beyond the percolation threshold. In addition, the maximum value of TMR_max_ in the (CoFeZr)_x_(MgF_2_)_100−x_ system (2.4%) is two times smaller than the corresponding value in the first, simpler system Co_x_(MgF_2_)_100−x_, (TMR_max_ = 5%), in which the paramagnetic phase of FeF_2_ is absent.

In our paper44 [45], we presented for the first time data on magnetic and magneto-optical (MO) properties and the effect of phase transitions on them when changing the composition of the composite. It was found that the appearance of the coercive force at the magnetic percolation threshold in (CoFeZr)_x_(MgF_2_)_100−x_ nanocomposites corresponds to the value x_FM_~34 at.% (Figure 9A,B), coincides with the electric percolation threshold x_per_, and both thresholds correspond to the formation of nanocrystals of the CoFeZr alloy of the hexagonal syngony.

Since, according to the value of the coercive force Hc, magnetic materials are divided into hard magnetic (Hc ≥ 50 Oe) and soft magnetic (Hc < 50 Oe), then the (CoFeZr)_x_(MgF_2_)_100−x_ nanocomposites with the maximum value Hc ≈ 30 Oe at x = 47 at.% should be classified as soft magnetic materials in accordance with the results presented in Figure 9C.

It should also be noted that the threefold decrease in the maximum value of the coercive force Hc ≈ 30 Oe and the significant “softening” of the magnetic properties of the (CoFeZr)_x_(MgF_2_)_100−x_ system compared to the magnetic hard Co_x_(MgF_2_)_100−x_ system may be due to the appearance of Fe-F chemical bonds at the interphase boundaries with the formation of the paramagnetic compound FeF_2_.

In addition, when studying nanocomposites using magneto-optical spectroscopy methods on the concentration dependences of the transverse Kerr effect (TKE) (Figure 9D), two maxima of the absolute values of the TKE were revealed, corresponding to the structural-phase transitions of the CoFeZr alloy. The first maximum corresponds to the transition from an X-ray amorphous to a nanocrystalline state with a hexagonal structure at x ≈ 30–34 at.%, and the second to the transition of CoFeZr nanocrystals from a hexagonal to a cubic body-centered structure at x ≈ 43 at.%.

Comparing the obtained results of complex studies of the atomic composition, structural-phase organization, and electromagnetic properties of two metal–insulator nanocomposite systems Co_x_(MgF_2_)_100−x_ and (CoFeZr)_x_(MgF_2_)_100−x_, we come to the conclusion that the well-known composition–structure–property triad manifests itself in these materials in its entirety.

In this case, the primary factor in this triad is the atomic composition of the metal component in the nanocomposite. In our case, this is a change in the atomic composition from one metallic cobalt Co in the dielectric matrix MgF_2_ in the first system to a three-element alloy CoFeZr in the same matrix of the second system. The two studied systems have in common the regularity that we recorded for the first time, which consists of the fact that at the first stage of self-organization of nanocomposites during ion-beam sputtering of composite targets on substrates in both systems, a heterophase material is formed, which contains one amorphous and one nanocrystalline phase. Which of the two parts, the metallic or the dielectric, forms nanocrystals, depends on the relative content of the metallic phase x in the composite and coincides with the percolation thresholds, both electric x_per_ (by the flow of TMS current) and magnetic x_FM_ (superparamagnetic-soft magnetic transitions).

In this case, the different atomic composition of the metal component of the two systems Co_x_(MgF_2_)_100−x_ and (CoFeZr)_x_(MgF_2_)_100−x_ affects the electromagnetic properties of the nanocomposites. In both systems, the superparamagnetic-soft magnetic transitions coincide with the formation of nanocrystals of the metal component at close concentrations: x = 34 at.% in the first system and x = 37 at.% in the second. However, with a further increase in x in the first system, the nanocomposites with a cobalt content of Co x > 42 at.% beyond the percolation threshold transition from a soft magnetic state to a hard magnetic state with a coercive force of up to 95 Oe, whereas in the second system, with an increase in the content of the CoFeZr alloy in the nanocomposites, soft magnets with a coercivity of no more than 30 Oe remain beyond the percolation threshold.

The general decrease in magnetism in a system with a three-element CoFeZr alloy can be understood using the basic idea of the Giaccarino–Waller phenomenological model [50,51], which is that in transition metal alloys, the magnitude of the local magnetic moment on a d-metal atom is determined primarily by the composition of the immediate environment.

Then, in [52], it was shown that in such a model, not only the magnitude but also the direction of the local magnetic moment depends on the immediate environment of the d-metals. Therefore, with an increase in the number of chemical bonds of d-metal atoms Co, Fe, and Zr with impurity oxygen atoms and boundary fluorine atoms from the dielectric matrix, the local magnetic moments on the Co and Fe alloy atoms can not only decrease but also flip opposite to the direction of the overall magnetization of the system. As a result, the magnetic properties of the heterogeneous multiphase nanocomposite system with CoFeZr alloy nanocrystals are significantly softened compared to nanocomposites containing cobalt nanocrystals in the same MgF_2_ dielectric matrix.

The next stage of our work is experimental studies of two nanocomposite systems at the atomic level. For these purposes, X-ray spectroscopy XANES (X-ray absorption near edge structure) on K-edges 3d metals Co and Fe was used.

## 4. XANES Spectra of the K-Edges of Co and Fe in Nanocomposites Co_x_(MgF_2_)_100−x_ and (CoFeZr)_x_(MgF_2_)_100−x_

X-ray absorption spectrometry is based on measuring the dependence of the X-ray absorption coefficient µ on the energy E of X-ray photons incident on the sample.

The linear coefficient of X-ray absorption is determined by the formula:(4)$$\mu =\ln\frac{I_{0}}{I_{t}}$$
where I_0_ is the intensity of the incident radiation, and I_t_ is the intensity of the radiation transmitted or reflected by the absorbing layer t of the sample.

As the energy of an X-ray photon approaches the binding energy of an electron at a deep internal level of an atom, the absorption coefficient µ(E) in the material increases sharply, and this jump is called the absorption edge. Depending on the principal quantum number n of the excited electron shell, the edges of X-ray absorption are designated by the Latin letters K (n = 1), L (n = 2), M (n = 3), N (n = 4), etc.

Near the absorption edge, the dependence µ(E) has a fine structure of an oscillating nature [53,54,55], which occurs due to the interference of the primary wave of the photoelectron with secondary waves that arise during its scattering by atoms of the nearest environment.

The X-ray absorption spectrum is conventionally divided into two regions [52]. One of them near the threshold of absorption is called XANES (X-ray absorption near-edge structure). The energy region it occupies extends from ~50 eV before the absorption edge to 100–150 eV after the edge. In the low-energy region of XANES, electrons have a large mean free path in the material, are elastically scattered by atoms of the nearest environment, and participate in multiple scattering. The participation of an electron in multiple scattering makes the XANES theory difficult to describe quantitatively, but even a qualitative analysis of the near-edge region of X-ray absorption spectra allows one to obtain information on the valence of the absorbing atom, the coordination environment, and its symmetry by comparing the experimental spectrum with reference spectra from known phases [54,55,56].

The second region is called EXAFS (extended X-ray absorption fine structure), which extends in the energy region above the absorption edge, i.e., beyond the absorption threshold by 500–1000 eV. This distribution is due to the fact that the mean free path of low-energy electrons is significantly higher, which allows them to travel long distances by undergoing multiple scattering, elastically reflecting from the electron shells of neighboring atoms, unlike high-energy electrons, which lose energy during the transfer of momentum through the Compton effect already at the first collision and participate in scattering once. The oscillatory structure of the EXAFS spectrum is formed due to the scattering of photoelectrons on atoms of the local environment. This method allows one to obtain an idea of the local environment of the absorbing atom, the number of neighboring atoms, as well as interatomic distances [54,55,56,57,58], i.e., the short-range order of the substance.

A qualitative analysis of Co/Fe-K XANES/EXAFS spectra in two close metal–insulator nanocomposite systems aims to show at the atomic level that changes in the atomic structure of a metal/alloy during the transition from an amorphous to a nanocrystalline state (without a change in valence) in the percolation threshold region affect the change in the oscillatory fine structure of the XANES and EXAFS spectra.

### 4.1. XANES Spectra of Co K-Edges in Co_x_(MgF_2_)_100−x_ Nanocomposites

In the system of nanocomposites with metallic cobalt Co in the dielectric matrix MgF_2_, the XANES spectra of Co K-edge were obtained for four samples with different metal concentrations, two samples from the region before the percolation threshold (x = 19, 27) and after the threshold (x = 42, 59).

Figure 10A shows the XANES spectra in the Co K-edge absorption region of cobalt in these samples together with the reference spectrum (Co 100%) from the α-Co metal foil.

Visual comparison of the oscillating structure of the XANES spectra from the nanocomposite samples and the reference spectrum of the microcrystalline α-Co foil (Figure 10A) shows that in two samples with the maximum metal concentration x = 59 and x = 42 (beyond the percolation threshold x_per_ = 37 at.%) and nanocrystal sizes of about 20 nm, it is similar to the XANES of the reference spectrum of Co (100%).

Therefore, the symmetry of the arrangement of cobalt atoms in nanocrystals in an amorphous MgF_2_ matrix is similar to the symmetry of the crystal structure of metallic α-Co with a hexagonal close packing of atoms and a coordination number close to 12.

While in two other samples with a lower concentration of amorphous metal clusters, x = 19 and x = 27 (up to the percolation threshold), the XANES spectra differ from the reference α-Co with a more intense and expressive first maximum at the main K-edge of absorption, indicating a possible change in the short-range order in nanocomposites with compositions below the percolation threshold.

The assumption of such a change was confirmed by analyzing the Fourier transform of an extended region of the experimental spectra beyond the Co K-edge of absorption, representing the radial distribution function of Co atoms without correction for the scattering phase shift (Figure 10B). Here, the main maximum of the radial distribution function of Co atoms decreases its intensity almost three times in samples with a lower metal concentration below the percolation threshold (x = 19 and 27) compared to the other two samples (x = 42 and 59) beyond the percolation threshold, indicating a decrease in the number of metal atoms in the immediate vicinity of cobalt in Co metal clusters immersed in the MgF_2_ nanocrystalline matrix.

Also, in the radial distribution curves up to the percolation threshold, there is no noticeable signal from the second coordination sphere at distances greater than 4 A, which additionally confirms the small size and/or strong disorder in the structure of small metallic cobalt clusters. Thus, in the Co_x_(MgF_2_)_100_ nanocomposite system, the concentration dependences of the structural-phase properties also manifested themselves at the atomic level of the local environment of Co atoms, which was determined based on the rearrangement of the fine structure of the X-ray absorption spectra in the region of the Co K-edges.

### 4.2. XANES Spectra of Co and Fe K-Edges in Nanocomposites (CoFeZr)_x_(MgF_2_)_100−x_

In the system of nanocomposites with a three-element alloy CoFeZr in a dielectric matrix MgF_2_, XANES spectra were obtained for six samples with different metal concentrations in the nanocomposites x = 20, 25, 30, 45, 50, 100, three samples from the region before the percolation threshold (x_per_ = 34 at.%) and three samples from the region after the percolation threshold, including a sample of a thin-film amorphous alloy without a dielectric (x = 100).

Figure 11 shows the XANES spectra of Fe and Co with the K-edge positions for the two metals of the same alloy aligned on the main absorption threshold in a single energy scale. The energy scale for the Fe K-edge is shown at the top, and the energy scale for the Co K-edge is shown at the bottom of the figure. The dotted curves indicate the reference spectra of the metal foil α-Fe with a body-centered cubic structure and a coordination number of 8 (upper dotted curve) and the metal foil of α-Co with a hexagonal close-packed structure and a coordination number of 12 (lower dotted curve), the fine oscillatory structure of which differs significantly due to the different symmetry and different short-range of their crystal structure.

In Figure 11A, all six XANES spectra with different x values for each of the two metals, Fe and Co, are combined in the normalized intensity scale of the absorption coefficient µ in order to demonstrate the almost complete similarity of the fine structure of the spectra in nanocomposites of any composition for both 3d metals, Fe and Co, located next to each other in the periodic table of elements. The similarity of the fine structure of the XANES K-spectra of the two metals indicates the equal participation of Co and Fe atoms in the formation of nanocrystals of the CoFeZr alloy of either of the two symmetries, hexagonal close-packed (up to the percolation threshold) or body-centered cubic (after the percolation threshold).

In Figure 11B, spectra with different alloy contents x are presented individually one above the other in the intensity scale to show how the fine structure of the XANES spectrum behind the K-absorption edges of the two metals simultaneously responds to a change in the symmetry of the nanocrystal structure from cobalt like to iron like during the phase transition of the nanocrystals from the hcp to the bcc structure (samples x = 45; 50).

In this system, the most structureless, but still similar, are the XANES K-spectra of two metals, Co and Fe, in a sample of pure CoFeZr alloy (x = 100), which is due to its amorphous state and the complete absence of long-range order, even limited by the size of the nanocrystals.

Figure 12 shows the Fourier transforms of the experimental spectra beyond the K-edge of cobalt absorption of (CoFeZr)_x_(MgF_2_)_100−x_ nanocomposites, representing the radial distribution function of Co atoms for nanocomposites with different alloy contents x without taking into account the phase shift during photoelectron scattering. Here the main maximum of the radial distribution function of Co atoms is located close to the Co-Fe interatomic distance in the alloy; its intensity decreases almost twofold in all samples compared to the reference samples of metal foil of both metals, indicating a significant change in the short-range order in metal clusters/nanocrystals in nanocomposites towards decreasing coordination numbers. However, the Fourier transforms of two samples with the highest alloy concentrations, x = 45 and x = 50, beyond the percolation threshold x_per_ = 34 at.% become similar to the Fourier transforms of α-Fe with three pronounced maxima in accordance with the phase transition of the alloy nanocrystals from the cobalt-like hcp structure (samples with x = 20; 25; 30) to the iron-like bcc structure in samples x = 45; 50 with maximum nanocrystal sizes of about 20 nm, determined from X-ray diffraction data presented in Figure 4.

Thus, the transformation of the local environment of two Co and Fe atoms of the magnetic alloy at the percolation threshold is manifested at the atomic level in the fine structure of the X-ray absorption spectra XANES at the K-edges of Co and Fe and in the Fourier transforms of the spectra beyond the Co K-edge absorption in the nanocomposite (CoFeZr)_x_(MgF_2_)_100−x_ system.

## 5. Conclusions

For the first time, a comparative analysis of a complex study of the atomic composition, structural and phase organization, electromagnetic properties, and XANES spectra of two nanocomposite systems Co_x_(MgF_2_)_100−x_ and (CoFeZr)_x_(MgF_2_)_100−x_ present.

XRD results show that for any composition, the nanocomposite consists of one X-ray amorphous phase and one nanocrystalline phase. Which of the two components, metallic or dielectric, forms nanocrystals depends on the relative content of the metallic component x, and it is determined by the percolation threshold x_per_ = 37 at.% in the first system and x_per_ = 34 at.% in the second system. The average size of metal nanocrystals in both systems varies within 10–20 nm.

The main role of zirconium atoms of the sputtered three-element alloy Co_45_Fe_45_Zr_10_ is to bind impurity oxygen atoms with the formation of stable chemical bonds Zr-O with the formation of impurities of ZrO_2_, detected by X-ray photoelectron spectroscopy (XPS) using ion etching of the surface layers of nanocomposites.

At the metal–dielectric interfaces in the (CoFeZr)_x_(MgF_2_)_100−x_ nanocomposites, the formation of Fe-F chemical bonds between iron and fluorine atoms with the formation of a paramagnetic FeF_2_ phase was detected by Mössbauer spectroscopy. That affects the reduction in magnetic properties in this system, compared to the Co_x_(MgF_2_)_100−x_ system.

The percolation thresholds in Co_x_(MgF_2_)_100−x_ nanocomposites at x_per_ = 37 at.% and in (CoFeZr)_x_(MgF_2_)_100−x_ nanocomposites at x_per_ = 34 at.%, determined from the concentration dependences of electrical resistance, coincide with the formation of metallic nanocrystals in the amorphous dielectric matrix of MgF_2_.

In the concentration dependences of the magneto-optical equatorial Kerr effect in the Co_x_(MgF_2_)_100−x_ system, one maximum absolute value of TKE is observed at the percolation threshold at x = 37 at.%, coinciding with the formation of α-Co nanocrystals of hexagonal syngony, whereas in the (CoFeZr)_x_(MgF_2_)_100−x_ system, two maxima appear, one of which (x ≈ 30–34 at.%) corresponds to the formation of nanocrystals of the CoFeZr alloy of hexagonal symmetry, and the second maximum at x ≈ 45 at.% corresponds to the phase transition of nanocrystals from a hexagonal structure to a cubic body-centered structure.

In the Co_x_(MgF_2_)_100−x_ system, the formation of hexagonal Co nanocrystals occurs in the region of electric and magnetic percolation thresholds at x_per_ ≈ 37 at.% and is accompanied by a transition of the system from the superparamagnetic to the ferromagnetic state. With an increase in the metal content (x > 42 at.%), this state acquires a magnetically hard character with a coercive force of Hc ≈ 95 Oe.

In the (CoFeZr)_x_(MgF_2_)_100−x_ system, the formation of hexagonal nanocrystals occurs in the region of electric and magnetic percolation thresholds at x_per_ ≈ 34 at.%. Below this value, the nanocomposites exhibit superparamagnetic properties, and at higher x values they become soft magnetically and retain this state far beyond the percolation threshold and after the phase transition of nanocrystals from the hcp to the bcc structure at x = 43 at.%, with a maximum value of the coercive force Hc ≤ 30 Oe. We associate such a decrease in magnetism compared to the first system with the formation of the paramagnetic phase FeF_2_ at the interface of the CoFeZr alloy nanocrystals with the MgF_2_ dielectric matrix.

The concentration dependences of the structural-phase properties of the two nanocomposite systems are manifested at the level of the local environment of Co and Fe atoms, which follows from the rearrangement of the fine structure of the XANES spectra in the region of percolation thresholds.

The similarity of the oscillating structure in the XANES K-spectra of two metals, Co and Fe, in (CoFeZr)_x_(MgF_2_)_100−x_ of any composition indicates the same local environment of Co and Fe atoms in the CoFeZr alloy at any symmetry of clusters and nanocrystals, hexagonal or cubic.

The oscillating structure of the XANES K-spectra of Co and Fe changes simultaneously and remains similar beyond the percolation threshold and after the phase transition from the hcp to the bcc structure at x = 43.

A significant decrease in the amplitude of the spectral oscillations in the deposited film of pure CoFeZr alloy (x = 100) is explained by its X-ray amorphous state and the absence of long-range order, even limited by the size of the nanocrystals.

## Figures and Tables

**Figure 1 nanomaterials-15-00463-f001:**
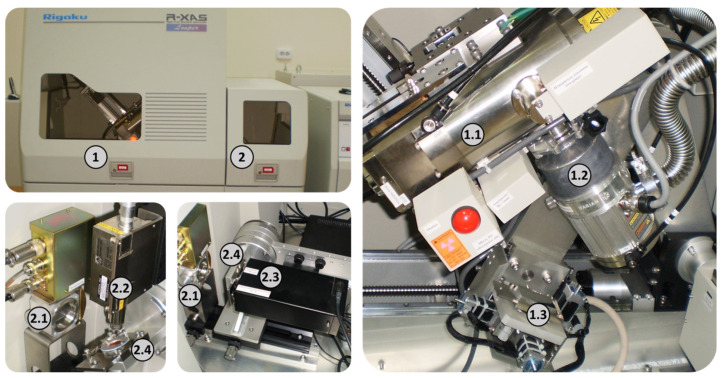
The main parts of the laboratory X-ray absorption spectrometer for measuring X-ray absorption spectra in transmission and fluorescence modes. Goniometer compartment 1 contains an X-ray tube 1.1, a turbomolecular pump 1.2 and a monochromator 1.3. These devices are mounted on a Rowland circle and direct monochromated radiation into the measuring chamber 2. Detector 2.1 measures the intensity of incoming radiation, while detector 2.2 or detector 2.3 measures the intensity of fluorescence or radiation passing through the sample, respectively. Sample 2.4 is mounted on a rotating holder, which is adjusted for both [measurement schemes].

**Figure 2 nanomaterials-15-00463-f002:**
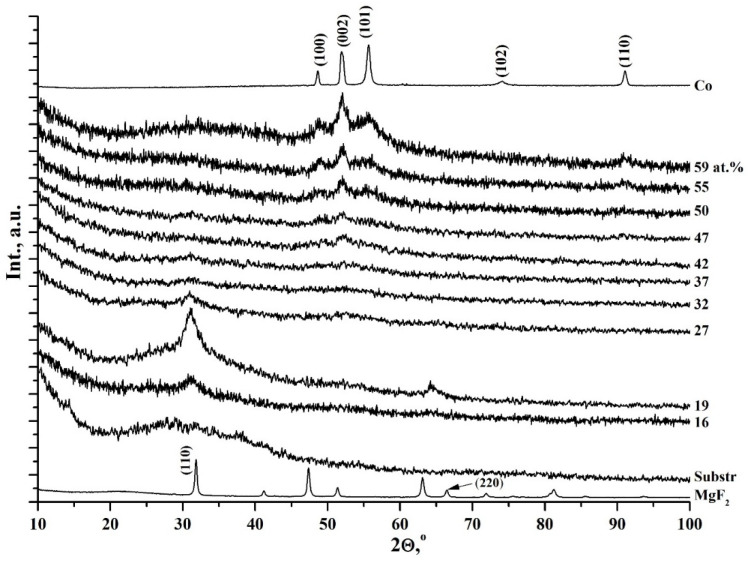
Diffractograms of Co_x_(MgF_2_)_100−x_ nanocomposite samples of different compositions on the glass substrates. The x values (in at.%) are indicated on the right of each diffractograms. The upper diffractogram belongs to metallic α-Co, and the lower one to polycrystalline MgF_2_ [41].

**Figure 3 nanomaterials-15-00463-f003:**
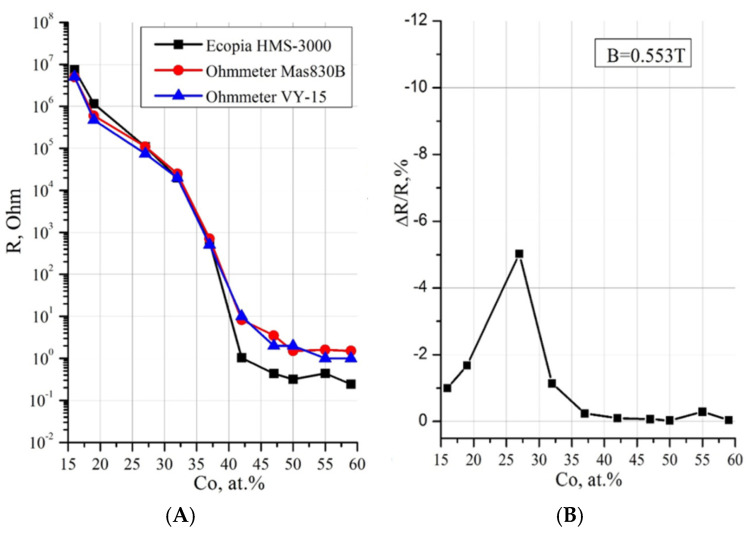
Concentration dependences of electrical resistance R (**A**) and tunnel magnetoresistance TMR (**B**) of Co_x_(MgF_2_)_100−x_ nanocomposites [42].

**Figure 4 nanomaterials-15-00463-f004:**
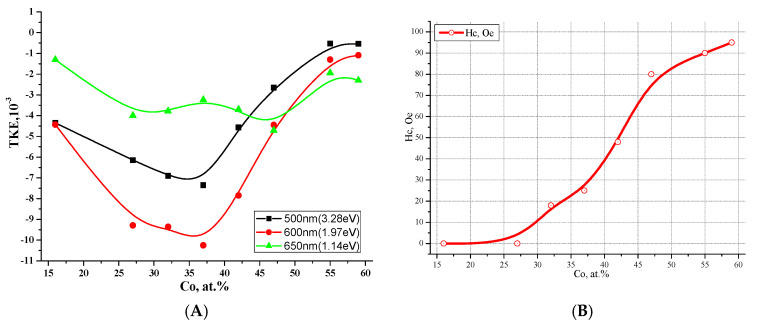
Concentration dependences in Co_x_(MgF_2_)_100−x_ nanocomposites: (**A**) transverse Kerr effect TKE at different values of incident light energy and (**B**) coercive force Hc [42].

**Figure 5 nanomaterials-15-00463-f005:**
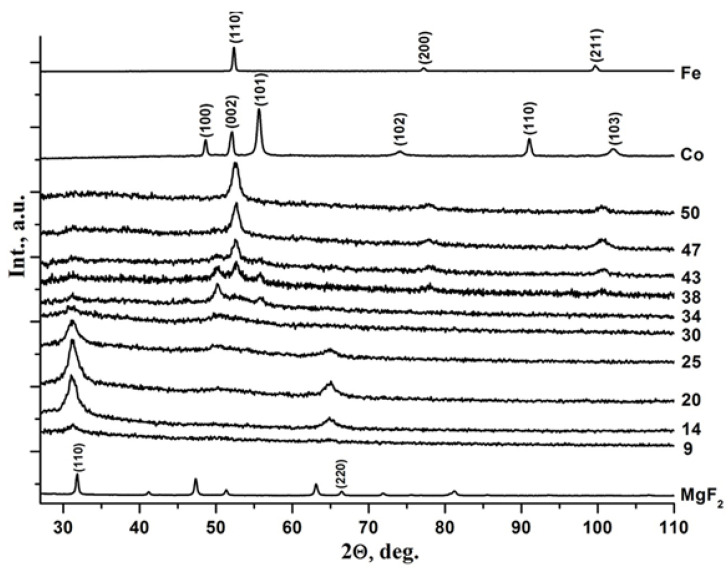
Diffractograms of (CoFeZr)_x_(MgF_2_)_100−x_ nanocomposite samples of different composition on glass substrates [45]. The values of x in at.% are indicated on the right of each diffractograms. The two upper diffractograms belong to metallic samples of α-Co and α-Fe, and the lower one to polycrystalline MgF_2_.

**Figure 6 nanomaterials-15-00463-f006:**
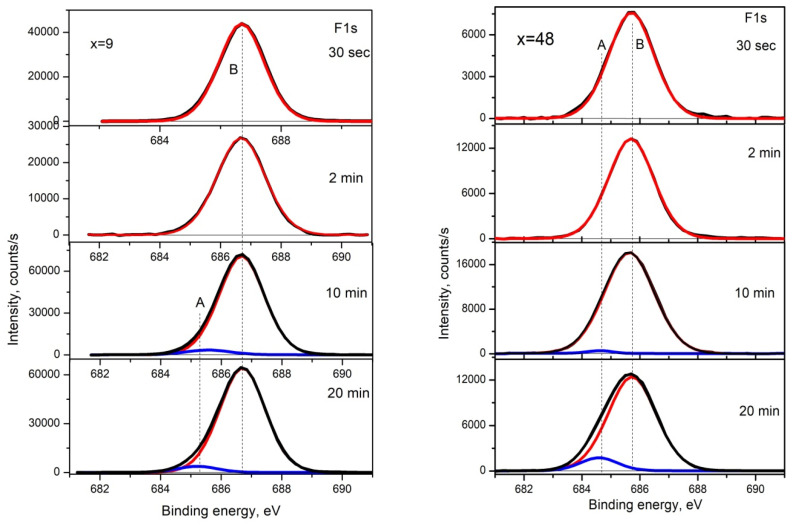
XPS spectra of fluorine F1s in the samples (CoFeZr)_x_(MgF_2_)_100−x_ with an alloy concentration of x = 9 (before the percolation threshold) and concentration of x = 48 (after the percolation threshold).

**Figure 7 nanomaterials-15-00463-f007:**
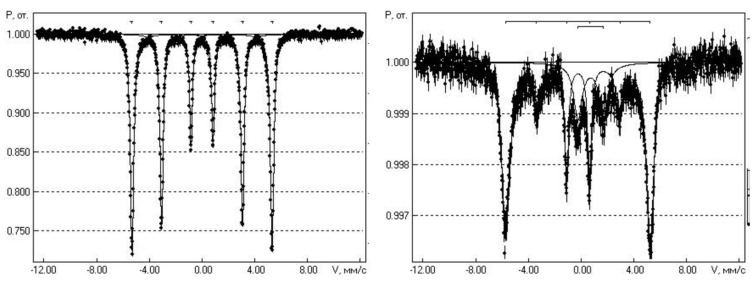
Mössbauer spectra from the calibration sample of α-Fe (**left**) and from the nanocomposite sample (CoFeZr)_51_(MgF_2_)_49_, measured at room temperature (**right**) [45].

**Figure 8 nanomaterials-15-00463-f008:**
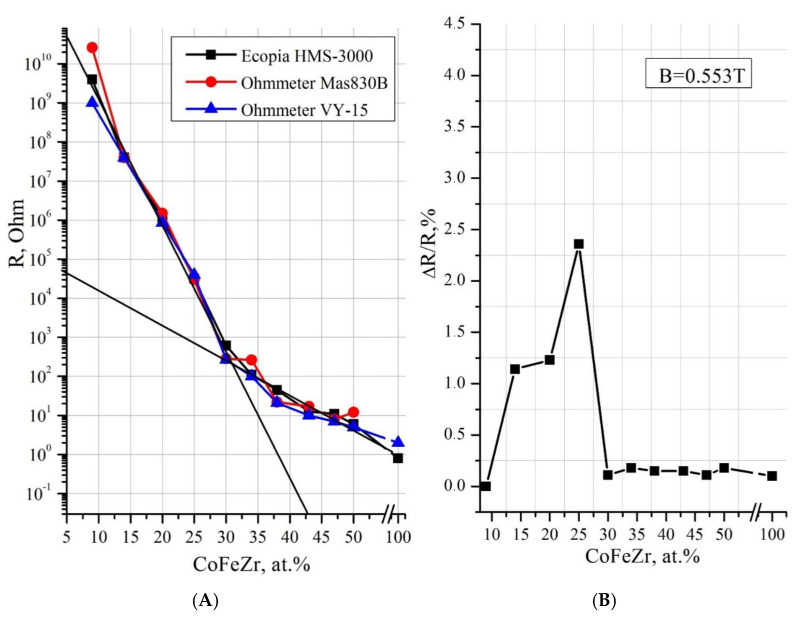
Concentration dependences of electrical resistance R (**A**) and tunnel magnetoresistance TMR (**B**) of (CoFeZr)_x_(MgF_2_)_100−x_ nanocomposites [45].

**Figure 9 nanomaterials-15-00463-f009:**
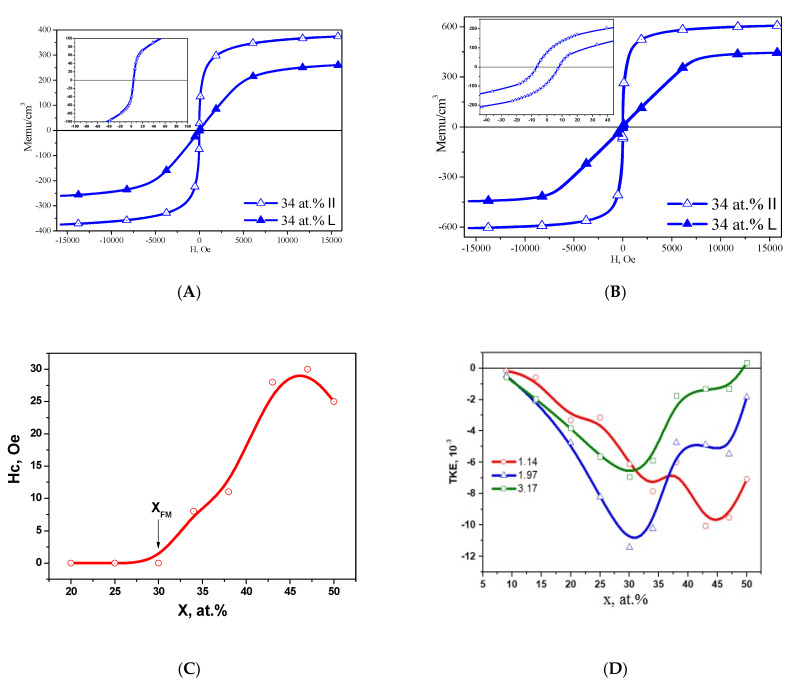
The absence of the hysteresis loop at x = 30 (**A**) and its appearance at x = 34 (**B**) in (CoFeZr)_x_(MgF_2_)_100−x_ nanocomposites. Concentration dependences of the coercive force Hc (**C**) and the transverse Kerr effect TKE at different values of the energy of light incident on the sample (**D**) [45].

**Figure 10 nanomaterials-15-00463-f010:**
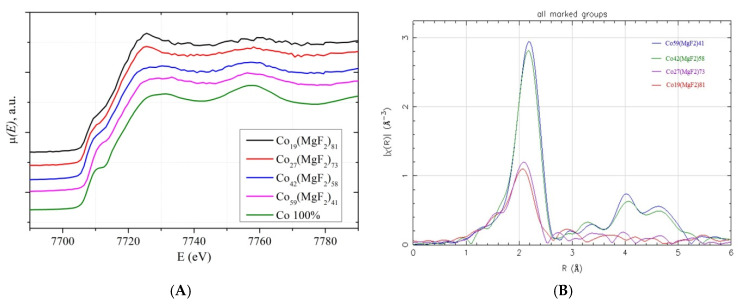
(**A**) XANES spectra in the region of the Co K-absorption edge and (**B**) Fourier transforms of the experimental spectra beyond the Co K-absorption edge in nanocomposites with different Co concentrations x.

**Figure 11 nanomaterials-15-00463-f011:**
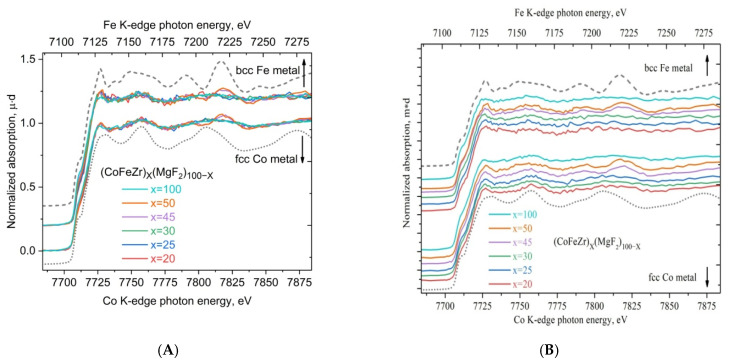
XANES spectra of Fe and Co with K-edges aligned in a single energy scale. The dotted curves denote the reference spectra of α-Fe and α-Co metal foils. In (**A**) all six XANES spectra with different x values are combined in the normalized intensity scale of the absorption coefficient µ for each of the two metals. In (**B**) the same XANES spectra with different x values are presented individually one above the other in the intensity scale.

**Figure 12 nanomaterials-15-00463-f012:**
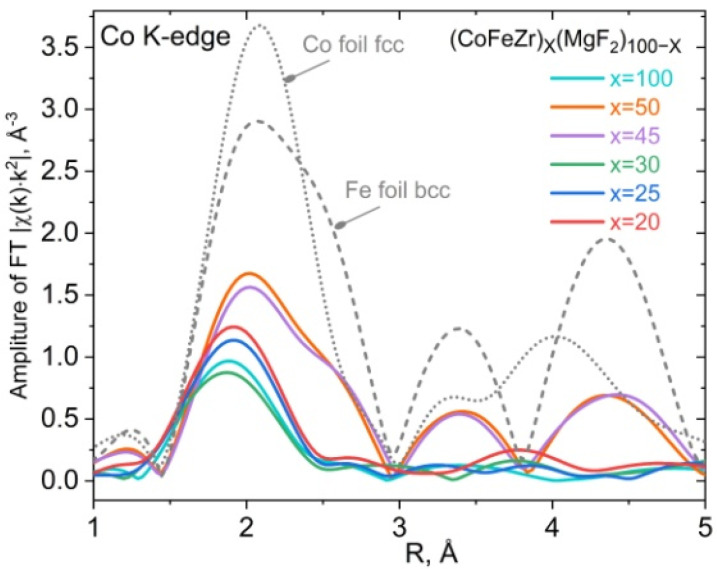
Fourier transform of the spectra behind the Co K-edge of absorption in samples with different alloy contents x in (CoFeZr)_x_(MgF_2_)_100−x_ nanocomposites. The dotted curves represent the reference samples of hcp α-Co and bcc α-Fe foils.

## Data Availability

Data available in a publicly accessible repository.

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
