# Peer review of "Feature of Nonlinear Electromagnetic Properties and Local Atomic Structure of Metals in Two Systems of Nanocomposites Cox(MgF2)100−x and (CoFeZr)x(MgF2)100−x"

_nanomaterials, 2025, doi:10.3390/nano15060463_

Round 1

Reviewer 1 Report

Comments and Suggestions for Authors

In this work, the authors reported the nonlinear electromagnetic properties and local atomic structure of metals in Cox(MgF2)100-x and (CoFeZr)x(MgF2)100-x nanocomposites. Various characterizations were used to reveal their findings. However, some important issues must be well solved before acceptance.

  1. The whole manuscript lacks the characterization on F element, which can be analyzed by XPS. Necessary characterizations should be conducted.
  2. For the analysis of Co/Fe-K EXAFS, the standards of Co and Fe metal should be compared with literature to support the reliability of these data. For example, the authors can refer to the paper with DOI of 10.1002/adma.202305074.
  3. Figures 8 and 9a are not clear, please improve the quality of the figures.
  4. The analysis processes on the Co/Fe-K XANES and EXAFS should be clearly provided, please refer to the paper with DOI of 10.1002/adma.202103392.
Comments on the Quality of English Language

The English could be improved to more clearly express the research.

Author Response

Dear reviewer 1! We thank you for the evaluation of our article and provide answers to your comments and an expanded version of the article taking into account your comments.

Response 1.We have added from our work [43] Figure 6 with XPS spectra of the element F, which weakens the magnetic properties of the nanocomposite system with CoFeZr alloy, with corresponding comments in the text:

As an example of such uncertainty, we present Figure 6 from our work [43], which shows the XPS spectra of fluorine F1s from one of the samples with the alloy concentration x=9 (before the percolation threshold) and from the sample with the alloy concentration x=48 (after the percolation threshold), obtained at different ion etching times from a minimum of 30 sec to a maximum of 20 min, after which the spectra practically do not change their shape. The results show that all XPS spectra of fluorine F1s are a single-component practically symmetric curve with a maximum binding energy of ~ 686.0 eV, corresponding to magnesium difluoride MgF2[48]. And only after deep etching for 10-20 minutes, a very weak component with a binding energy of about 685.0 eV appears in the spectra, which can relate to the boundary bonds of fluorine with the any atoms of d-metals Co, Fe and Zr[48].

Response 2.Following the reviewer's recommendations, we have added two references [53,54] to confirm the reliability of our Co/Fe-K EXAFS data.                   

[53] Daqin Guan, Hengyue Xu, Qingwen Zhang et all., Identifying a Universal Activity Descriptor and a Unifying Mechanism Concept on Perovskite Oxides for Green Hydrogen Production Advanced Materials, (2023),35(44), 2305074      DOI 10.1002/adma.202305074.

[54] Daqin Guan, Kaifeng Zhang, Zhiwei Hu et all, Exceptionally Robust Face-Sharing Motifs Enable Efficient and Durable Water Oxidation, Advanced Materials, (2021),33(41),2103392, DOI 10.1002/adma.202103392.

Response 3. We tried to improve the quality of the pictures.

Response 4. We have added the Daqin Guan papers [54,55] you recommended and text:«The qualitative analysis of Co/Fe-K XANES and EXAFS spectra in two closel metal-insulator nanocomposite systems aims to show at the atomic level that changesb  in the atomic structure of the metal/alloy during the transition from the amorphous to the nanocrystalline state (without a change in valence) in the percolation threshold region affects the change in the oscillatory fine structure of the XANES and EXAFS spectra.»

We believe that the qualitative analysis of the spectra and their Fourier transforms presented in Figures 10–12 achieves this goal and shows the change in the oscillatory fine structure of the XANES and EXAFS spectra in samples with metal/alloy concentration x (at.%) before and after the percolation threshold in both systems. Nevertheless, we found it useful to include in the text of the paper the references suggested by the reviewer.

Reviewer 2 Report

Comments and Suggestions for Authors

This study provides comprehensive experimental insights into the structural, electrical, magnetic, and spectroscopic properties of nanocomposites. The correlation between percolation thresholds, nanocrystal formation, and phase transitions is well-established, and the use of multiple characterization techniques (XANES, XPS, Mössbauer spectroscopy, and the Kerr effect) effectively supports the conclusions. However, several aspects could be improved to strengthen the manuscript:

  1. The author should provide more detailed information about the samples, particularly the layer structure used for TMR measurements and the magnetic field dependence of TMR.
  2. A brief comparison with similar metal-dielectric systems would enhance the study's context and highlight its broader significance.
  3. The author should emphasize the significance and value of their research more clearly in the abstract, introduction, and conclusion sections to better convey its impact.

Addressing these points would further improve the clarity and impact of the manuscript.

Author Response

Dear reviewer 2! We thank you for the evaluation of our article and provide answers to your comments and an expanded version of the article taking into account your comments.

Response 1: We presented in Section 3.1 and 3.2 more detailed information on the layer structure in the samples for the variation of TMR depending on the metal/alloy concentration and compared the maximum value of TMRmax with the corresponding values ​​in cobalt composites with other oxide matrices:

In Section 3.1 : To obtain this dependence, the value of TMS=ΔR/R(0) in accordance with relationship (1.1) was measured in a magnetic field with induction B = 5.5 kGs on 10 samples with different values ​​of cobalt concentration, of which 5 samples up to the percolation threshold and at the maximum TMS contained amorphous cobalt in the nanocrystalline matrix MgF2, and 5 other samples beyond the percolation threshold x≥37 contained hexagonal cobalt nanocrystals a-Co in the amorphous matrix MgF2. At a cobalt concentration of x=27 at.%, up to the percolation threshold, there is a sharp increase in TMS to the maximum value TMSmax=ΔR/R(0) =5%. It significantly exceeds the corresponding value of TMCmax in cobalt nanocomposites with other oxide matrices: in the Cox(Al2O3)100-x system TMCmax≈ 1.4%, and in the Cox(SiO2)100-x system TMCmax≈0.5% in a magnetic field with an induction of 10 kGs [10].

In Section 3.2: Figure 8B reproduces the concentration dependence of the tunnel magnetoresistance TMR=ΔR/R(0) in accordance with relation (1.1), which was measured in a magnetic field with induction B = 5.5 kGs on 11 samples with different concentrations of the CoFeZr alloy, of which 5 samples up to the percolation threshold and at the maximum TMR contained an amorphous alloy in a nanocrystalline MgF2 matrix, and 6 other samples beyond the percolation threshold x≥34 at.% contained hexagonal nanocrystals of the alloy in an amorphous MgF2 matrix.

Response 2:  At an alloy concentration of x=25 at.%, up to the percolation threshold, a sharp increase in TMR occurs to the maximum value TMRmax=ΔR/R(0) =2.4%. This value is less than the corresponding value of TMRmax in a composites of the same alloy in an oxide matrix (CoFeZr)x(Al2O3)100-x with a value of TMCmax≈3.5% in a magnetic field with an induction of 10 kGs [10].

Response 3:  We have tried to emphasize the novelty of the results obtained in the abstract, introduction and conclusion.

Reviewer 3 Report

Comments and Suggestions for Authors

Some comments are provided as follows:

  1. Scale bars should be added in Figure 1.
  2. Necessary electronic structural characterizations should be added, e.g., XPS.
  3. All the figures are not clear, please improve them.

Author Response

Dear reviewer3! We thank you for the evaluation of our article and provide answers to your comments and an expanded version of the article taking into account your comments.

Response 1. Scale bars are usually shown on nano- and microscopic sized objects in SEM, TEM, AFM images. Whereas in Fig. 1 we show a general view of two main parts of the serial Japanese laboratory X-ray absorption spectrometer Rigaku R-XAS (upper left part of Fig. 1): part 1 with an X-ray source and part 2 with a detector of characteristic radiation reflected from the sample. The lower left part of Fig. 1 reproduces the internal content of part 1, the right part - the content of part 2 with an increase of about 5 times compared to the general view. This is obvious to experimenters dealing with such laboratory devices.

Response 2. We have added from our work [43] Figure 6 with XPS spectra of the element F, which weakens the magnetic properties of the nanocomposite system with CoFeZr alloy, with corresponding comments in the text:

As an example of such uncertainty, we present Figure 6 from our work [43], which shows the XPS spectra of fluorine F1s from one of the samples with the alloy concentration x=9 (before the percolation threshold) and from the sample with the alloy concentration x=48 (after the percolation threshold), obtained at different ion etching times from a minimum of 30 sec to a maximum of 20 min, after which the spectra practically do not change their shape. The results show that all XPS spectra of fluorine F1s are a single-component practically symmetric curve with a maximum binding energy of ~ 686.0 eV, corresponding to magnesium difluoride MgF2[48]. And only after deep etching for 10-20 minutes, a very weak component with a binding energy of about 685.0 eV appears in the spectra, which can relate to the boundary bonds of fluorine with the any atoms of d-metals Co, Fe and Zr[48].

Response 3. We tried to improve the quality of the pictures.